# Assessment of the Energetical and Biological Characteristics of Municipal Solid Waste from One of the Largest Landfills in Kazakhstan

Arlan Z. Abilmagzhanov ⬤, Nikolay S. Ivanov *⬤, Oleg S. Kholkin and Iskander E. Adelbaev

"DV Sokolskiy Institute of Fuel Catalysis and Electrochemistry" JSC, Kunaev Str. 142, Almaty 050010, Kazakhstan
* Correspondence: xumuk777@mail.ru

**Abstract:** Solid waste management is one of the main problems in large cities. The determination of the quantitative and qualitative composition of municipal solid waste is necessary for proper planning in the processing of such waste. This article presents the results of studies of the morphological composition, physico-chemical parameters and energetical potential of municipal solid waste (MSW) from the landfill of the city of Shymkent. Waste samples were taken from 10 different points of the landfill. The volume–mass parameters of the samples and their average density were determined. The combined waste samples mainly consisted of food waste, paper and cardboard, polymers and plastic, glass, metal, textiles, wood, leather, bones and rubber. Most of the waste, more than 30%, contains plastic products. The moisture of the combustible fraction of the samples is low and varies from 0.3% to 2.3%. The average ash content of the combustible fraction of the samples was determined and its average value is 18.4%. The yield of the volatile substances was determined and the average value per dry state is 78.7%. To determine the energetical potential of municipal solid waste, the gross calorific value was determined and the net calorific value was calculated, the average value of which is 24.15 MJ/kg. This shows the possibility of using solid waste as an energy source.

**Keywords:** municipal solid waste; garbage landfill; incineration; recycling; processing; fuel; ecology





## 1. Introduction

In recent years, an increasing amount of attention has been paid to the development of "green" technologies for the processing and disposal of municipal solid waste for the purpose of reuse and energy production, not only in developed countries, but also in Asian countries and the Middle East [1–4]. A clear understanding of the finiteness of natural resources and the fact that municipal solid waste is an inexhaustible and predictable source of secondary raw materials and fuel, comparable in energy efficiency to its traditional types, makes the development of such technologies necessary.

According to the US Environmental Protection Agency, there are 75 factories in the country that process municipal solid waste and extract energy by incinerating it. The combustible fraction after sorting allows for the generation of approximately 550 kWh of energy per ton of waste. At an average price of 0.04 USD per kWh, the profit per ton of solid waste is often between 20 and 30 USD [5].

The flagship among European countries in the field of waste-to-energy conversion is Sweden, which not only recycles almost all of its garbage, but also exports it from Norway, Ireland and the UK. According to the local waste management association "AvfallSverige", 99% of municipal waste is recycled in Sweden [6,7]. At the same time, 50.6% of waste is recycled, 48.6% is burned for energy production and only 0.8% is sent to landfills.

Kazakhstan adopted the Strategy "Kazakhstan-2050: a new political course of an established state", which defines clear guidelines for building a sustainable and efficient model of the economy based on the country's transition to a "green" path of development.

According to the concept of the transition to a "green economy", the state sets itself the task of increasing the share of recycling and disposal to 40% by 2030.

Despite the measures taken in terms of legislation and real actions, there is still no information about the characteristics of accumulated and newly incoming municipal solid waste at Kazakhstani landfills. The relevance of this work lies in obtaining reliable information on the properties, energy and ecological parameters of MSW.

The territory of Kazakhstan is located in four climatic zones, which differ significantly from each other in terms of the average temperature in winter and summer. Within the project implementation requirements, we covered all the climatic zones, in particular, Shymkent, which is the most southern city, Almaty, which is located 100 km to the north, Atyrau, which is 700 km to the north, and Astana, which is 1000 km to the north.

The second reason for choosing this city is that it is the republican significance city with a population of more than 1 million people. Further studies are planned for the cities of Astana, Almaty and Atyrau. These cities are located in the first three places in terms of quality and the standard of living, whilst Shymkent is in 12th place. Thus, we have made an attempt to assess the impact of the climate and living standards on the composition of MSW generated in these cities.

Taking all of the above into account and aiming to lay the foundation for policy development in the area of municipal solid waste treatment, the purpose of this article is to quantify the energetical potential of waste and environmental risks in one of the largest cities in Kazakhstan. For evaluation, the standard methods of analysis and research were used.

## 2. Results

Table 1 shows the volume–mass parameters of the MSW samples. The average density is 145.4 kg/m$^3$. In comparison, a group of Iraqi scientists cite density data equal to 229.088 kg/m3 for the city of Tikrit [8]; for India, a range from 300 to 400 kg/m$^3$ is typical for the city of Bhopal [9]; and for the capital of Oman, the given data correspond to 311.73 kg/m$^3$ [10]. On average, the MSW density values for industrialized countries such as the US and UK range from 100 to 150 kg/m$^3$, while for middle-income countries, this value is 175–330 kg/m$^3$; for countries with a low-income level (Bangladesh, Pakistan, Nepal, etc.), the density value is estimated to be 300–600 kg/m$^3$ [11,12]. A certain value of MSW density does not quite correspond to its global connection with the standards of living. The understated values may be due to the fact that the sampling was carried out in settled warm weather, as a result of which the samples contain an understated amount of moisture, which can significantly affect the density. For example, the most hygroscopic constituents of MSW, municipal residues, paper and textile materials practically did not contain any moisture.

**Table 1.** Volumetric and mass parameters of MSW samples.

| Parameter | Sample Number | | | | | | | | | |
|---|---|---|---|---|---|---|---|---|---|---|
| | 1 | 2 | 3 | 4 | 5 | 6 | 7 | 8 | 9 | 10 |
| Volume, L | 32 | 34 | 54 | 56 | 45 | 51 | 52 | 48 | 57 | 53 |
| Mass, kg | 5.47 | 4.967 | 5.399 | 7.943 | 5.889 | 7.98 | 7.36 | 8.514 | 8.569 | 7.344 |
| Density, kg/m$^3$ | 170.9 | 146.1 | 100.0 | 141.8 | 130.9 | 156.5 | 141.5 | 177.4 | 150.3 | 138.5 |

Table 2 shows the species composition of the MSW samples. The obtained data correlate with the world average values for the content of metals and glass, though there is an upward bias in the proportion of plastic and a downward bias in the proportion of food waste [13]. Paper waste is represented by the remains of printed materials (books, newspapers and magazines), packaging materials (boxes and paper containers) and paper products used for sanitary purposes (toilet paper, paper towels and wipes).

**Table 2.** Species analysis of MSW samples.

| Type of Waste, %wt. | Sample Number | | | | | | | | | |
|---|---|---|---|---|---|---|---|---|---|---|
| | 1 | 2 | 3 | 4 | 5 | 6 | 7 | 8 | 9 | 10 |
| Food waste | 10.7 | 32.0 | 0.0 | 0.0 | 0.0 | 17.2 | 0.0 | 8.8 | 20.2 | 22.7 |
| Paper and cardboard | 5.5 | 6.1 | 13.4 | 7.9 | 0.0 | 5.7 | 0.7 | 0.0 | 14.0 | 13.2 |
| Polymers | 49.8 | 12.3 | 33.0 | 42.1 | 34.2 | 26.4 | 39.6 | 24.7 | 14.2 | 29.6 |
| Glass | 0.0 | 5.8 | 7.1 | 2.7 | 0.0 | 0.0 | 1.8 | 6.5 | 0.0 | 0 |
| Ferrous metals | 1.6 | 0.0 | 0.0 | 15.4 | 26.5 | 10.4 | 0.2 | 7.8 | 0.7 | 0.3 |
| Non-ferrous metals | 4.0 | 2.8 | 3.8 | 0.6 | 0.6 | 0.3 | 6.2 | 0.0 | 7.2 | 0 |
| Textiles | 9.3 | 21.4 | 0.0 | 11.8 | 20.4 | 8.2 | 22.9 | 39.7 | 20.1 | 8.9 |
| Wood | 0.0 | 0.0 | 0.0 | 0.0 | 0.0 | 0.0 | 1.2 | 0.0 | 0.0 | 0 |
| Hazardous waste | 0.0 | 0.0 | 0.0 | 0.0 | 0.0 | 2.1 | 0.0 | 0.0 | 0.0 | 0 |
| Leather, bones, rubber | 8.9 | 0.0 | 16.3 | 12.5 | 1.1 | 19.7 | 11.7 | 3.1 | 9.4 | 2.5 |
| Municipal waste residue after removal of all other components | 10.2 | 19.7 | 26.4 | 7.1 | 17.2 | 10.1 | 15.7 | 9.3 | 14.2 | 22.8 |
| Total, % | 100 | 100 | 100 | 100 | 100 | 100 | 100 | 100 | 100 | 100 |

Polymer waste is represented by the widest range of products. This type of waste is made up of plastic bags and bottles, food containers, plastic scrap from household items, polypropylene construction bags and diapers.

Glass waste is represented by a small number of whole bottles, the main part of this type of waste is broken glass and damaged bottles, as well as containers from pharmaceutical preparations.

Ferrous and non-ferrous metals are represented by cans, fittings, other small construction waste and a small number of wires from electrical appliances.

Textiles are mainly represented by the remnants of clothing, footwear and cleaning products.

Wood waste consists of the remnants of furniture, tree branches and building materials. Used batteries constitute hazardous waste.

The skull of a cow was found in the sixth sample, and bones were also found in the fourth and ninth samples. The rest of the municipal waste contained the mineral part of sand, crushed stone, soil and tree leaves.

After sorting the samples, a combustible fraction was formed from paper, polymer, textile and wood waste. According to Table 3, the total content of combustible waste is 54.02%, more than half of which is plastic waste. Figure 1 shows the general morphological composition.

**Table 3.** The percentage composition of the combustible fraction of MSW.

| | Paper and Cardboard | Polymers | Textiles | Wood | Total |
|---|---|---|---|---|---|
| Total mass, g | 4628 | 20,988 | 11,808 | 88 | 37,512 |
| % of total mass of MSW | 6.66 | 30.22 | 17.00 | 0.12 | 54.02% |
| % of total combustible mass of MSW | 12.34 | 55.95 | 31.47 | 0.23 | 100% |

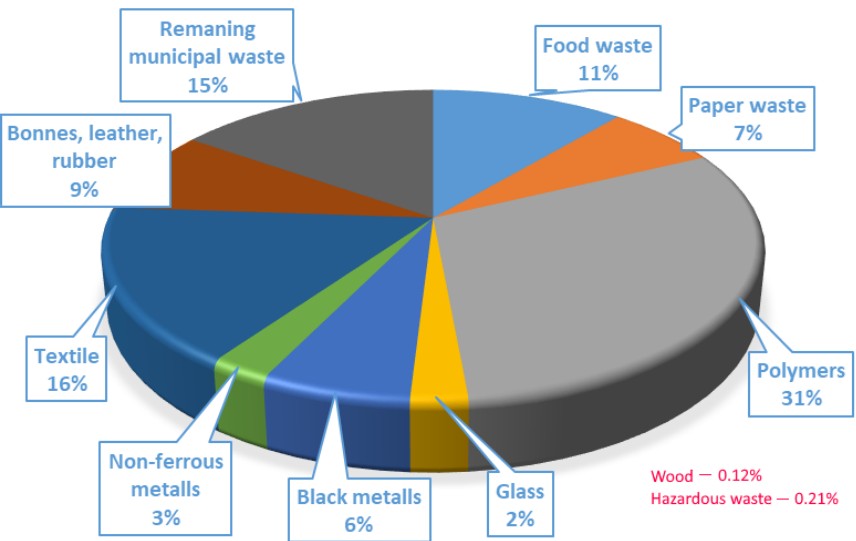

**Figure 1.** General morphological composition of MSW.

The sample moisture content was determined in two samples according to ISO 21660-3:2021 [14]. Table 4 shows the data on the determination of moisture. According to visual determination, the source of moisture in the samples was paper and textile materials, as well as soil particles adhering to the materials of the combustible fraction. The humidity of the samples was in the range of 0.3–2.3%, and the average humidity was 1.51%. These indicators can be considered very good, since the moisture in the fuel is ballast and the energy released during combustion will be spent on the evaporation of water [15].

**Table 4.** Moisture of samples.

| № of Sample | 1 | 2 | 3 | 4 | 5 | 6 | 7 | 8 | 9 | 10 | $W^a$ Avg. Total *, % |
|---|---|---|---|---|---|---|---|---|---|---|---|
| $W^a 1$, % | 1.7 | 1.3 | 1.7 | 1.2 | 0.3 | 1.8 | 0.6 | 1.5 | 2.1 | 1.9 | |
| $W^a 2$, % | 1.5 | 1.5 | 2.2 | 1.8 | 0.4 | 2.0 | 0.8 | 1.7 | 2.3 | 1.8 | 1.51 |
| $W^a$ avg., % | 1.6 | 1.4 | 2.0 | 1.5 | 0.4 | 1.9 | 0.7 | 1.6 | 1.7 | 1.9 | |

* $W^a$—humidity of samples.

The ash content was determined according to ISO 21656:2021 [16]. Table 5 shows the data on the determination of the ash content. The ash content of the samples was in the range of 5.9–33.3%, and the average ash content was 18.4%.

**Table 5.** Ash content of samples.

| № | 1 | 2 | 3 | 4 | 5 | 6 | 7 | 8 | 9 | 10 | $A^d$ Avg. Total *, % |
|---|---|---|---|---|---|---|---|---|---|---|---|
| $A^d 1$, % | 9.4 | 12.7 | 5.9 | 17.2 | 28.6 | 27.9 | 19.6 | 33.3 | 10.0 | 19.8 | |
| $A^d 2$, % | 9.3 | 12.3 | 5.8 | 17.0 | 28.6 | 27.8 | 19.4 | 33.2 | 9.9 | 19.7 | 18.4 |
| $A^d$ avg., % | 9.4 | 12.5 | 5.9 | 17.1 | 28.6 | 27.8 | 19.5 | 33.2 | 10.0 | 19.8 | |

* $A^d$—ash content of samples.

The yield of volatile substances was determined according to the ISO 22167:2021 standard [17]. Table 6 shows the data on the yield of volatile substances in a dry state, with an average value of 78.7%. The high yield of volatile substances and low ash content are related with the predominance of plastic in the samples [18,19].

**Table 6.** Yield of volatile substances from solid waste samples in a dry state.

| № | 1 | 2 | 3 | 4 | 5 | 6 | 7 | 8 | 9 | 10 | V$^d$ Avg. Total *, % |
|---|---|---|---|---|---|---|---|---|---|---|---|
| V$^d$1, % | 87.7 | 82.6 | 87.6 | 77.6 | 83.6 | 58.9 | 81.8 | 63.4 | 83.4 | 80.8 | |
| V$^d$2, % | 87.5 | 82.3 | 87.5 | 78.0 | 82.8 | 58.6 | 81.7 | 63.7 | 83.1 | 80.8 | 78.7 |
| V$^d$avg., % | 87.6 | 82.4 | 87.5 | 77.8 | 83.2 | 58.8 | 81.7 | 63.6 | 83.3 | 80.8 | |

* V$^d$—the yield of volatile substances from solid waste samples per dry state.

The gross calorific value was determined according to the ISO 1928:2009 *Solid Mineral Fuels—Determination of Gross Calorific Value by the Bomb Calorimetric Method and Calculation of Net Calorific Value* on an automatic GDY-1A+ isoperibol calorimeter. Table 7 shows the values of the gross calorific value considering and not considering the humidity of the samples. The obtained results show that the combustible fraction of MSW is quite suitable as a fuel in the cement industry [20].

**Table 7.** Gross calorific value of MSW samples.

| № | 1 | 2 | 3 | 4 | 5 | 6 | 7 | 8 | 9 | 10 |
|---|---|---|---|---|---|---|---|---|---|---|
| $Q^a_{s,V}$, mJ/kg | 26.82 | 23.00 | 23.92 | 28.79 | 24.16 | 23.24 | 26.54 | 19.17 | 29.82 | 27.34 |
| $Q^d_{s,V}$, mJ/kg | 27.26 | 23.32 | 24.40 | 29.24 | 24.25 | 23.68 | 26.72 | 19.49 | 30.34 | 27.86 |

$Q^a_{s,V}$—gross calorific value excluding humidity, $Q^d_{s,V}$—gross calorific value per dry state.

To calculate the net calorific value, the average hydrogen content in MSW was taken to be 7% [16] from the information presented by different countries around the world, and the average value was 24.15 mJ/kg.

Preliminary decomposition was carried out in accordance with GOST 55130-2012 "Solid fuel from municipal solid waste. Definition of macro elements".

The average content of the elements in the combustible fraction is as follows, presented as a percentage: Na—0.16; Mg—0.36; Al—1.22; Si—3.67; P—0.04; K—0.38; Ca—2.02; Mn—0.023; and Fe—0.7. Scanning microscopy was used to study the residue after ashing the average sample of the combustible fraction. Figure 2 shows the corresponding micrographs, which show that the sample contains both large particles with a size of 20–50 μm and very small ones, with a size of 1 μm or less. The particle size and composition of the ash correlates with the data obtained in [21].

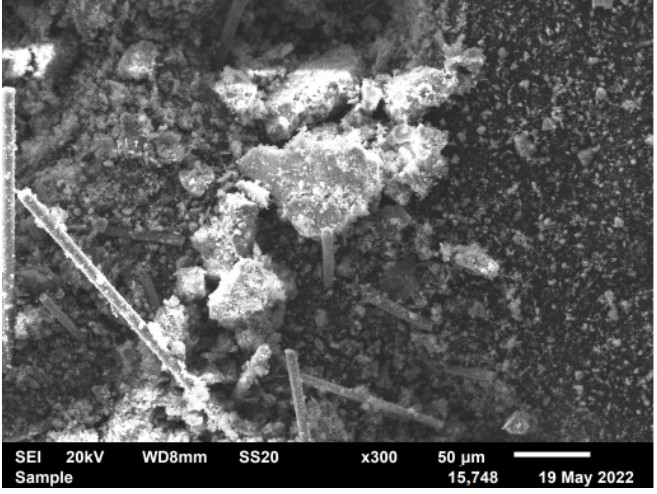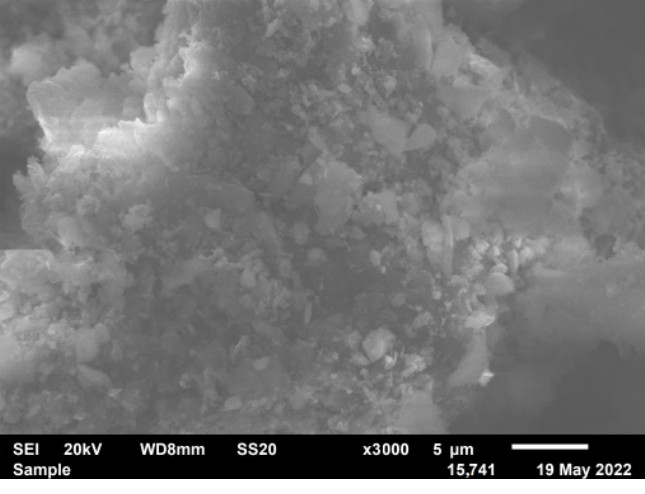

**Figure 2.** Micrographs of an ash sample.

To evaluate the biological hazard, a microbiological analysis of the water extracts from point samples 2, 4 and 6 was carried out. The following groups of microorganisms

were studied: heterotrophic bacteria, actinomycetes, microscopic fungi and bacteria of the *Escherichia coli* group. In addition to the risk of infection, it is necessary to take into account the risk of explosion when storing household waste, since bacteria capable of fermenting organic components are involved in processes of the formation of combustible gases [22].

According to Table 8, the results of the microbiological analysis of water extracts showed the presence of heterotrophic bacteria in all the samples under study in a significant amount. The greatest number was found in samples 4 and 6, where hundreds of millions of cells per 1 mL were detected. In sample 2, the number of bacteria was significantly (an order of magnitude) lower.

**Table 8.** The number of studied groups of microorganisms in samples of water extracts.

| Sample | The Number of Microorganisms, Colony Forming Units (CFU)/mL | | | | |
| | | | Microscopic Fungi | | |
| | Bacteria | Actinomycetes | Filamentous Mushrooms | Yeast | Bacteria of the *Escherichia coli* Group |
| --- | --- | --- | --- | --- | --- |
| 2 | $(6.95 \pm 1.2) \times 10^7$ | units | $(6.5 \pm 0.7) \times 10^3$ | not detected | $(4.15 \pm 0.9) \times 10^7$ |
| 4 | $(2.2 \pm 0.2) \times 10^8$ | not detected | $(1.35 \pm 0.7) \times 10^4$ | $(8.05 \pm 0.2) \times 10^5$ | $(6.15 \pm 0.9) \times 10^7$ |
| 6 | $(2.09 \pm 0.2) \times 10^8$ | not detected | $(6.5 \pm 0.7) \times 10^3$ | $(4.85 \pm 0.3) \times 10^4$ | $(2.55 \pm 0.3) \times 10^7$ |

Actinomycetes were found only in sample 2 in a single quantity.

Mycelial fungi were found in all the studied samples of water extracts. In total, 6500 cells were found in samples 2 and 6 per 1 mL, and in sample 4, this amount was twice as high. Yeast microflora was also found in samples 4 and 6. At the same time, their number exceeded the number of micromycetes by an order of magnitude and amounted to 805 thousand and 48.5 thousand CFU/mL, respectively.

Bacteria of the *Escherichia coli* group were also found in all the studied samples of water extracts. On the Chromagar Orientation medium, tens of millions of cells were found per 1 mL. The largest number was noted in sample 4. The determination of the bacteria of the *Escherichia coli* group using the fermentation method on a lactose-peptone medium, followed by inoculation on the Endo medium, showed that the coli index of these samples was more than 1100, i.e., there are more than 1100 cells in 1 L of water. Figures 3–5 show the images of cultured microorganisms from the studied extracts.

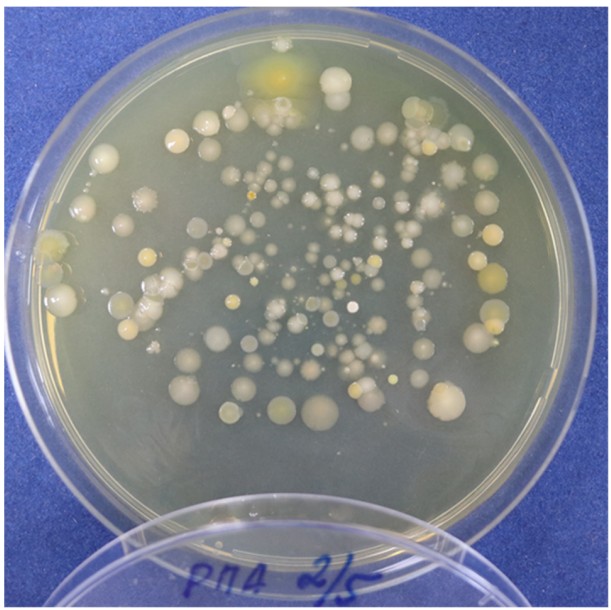

**Figure 3.** Colonies of heterotrophic bacteria from a sample of water extract (dilution $1:10^5$).

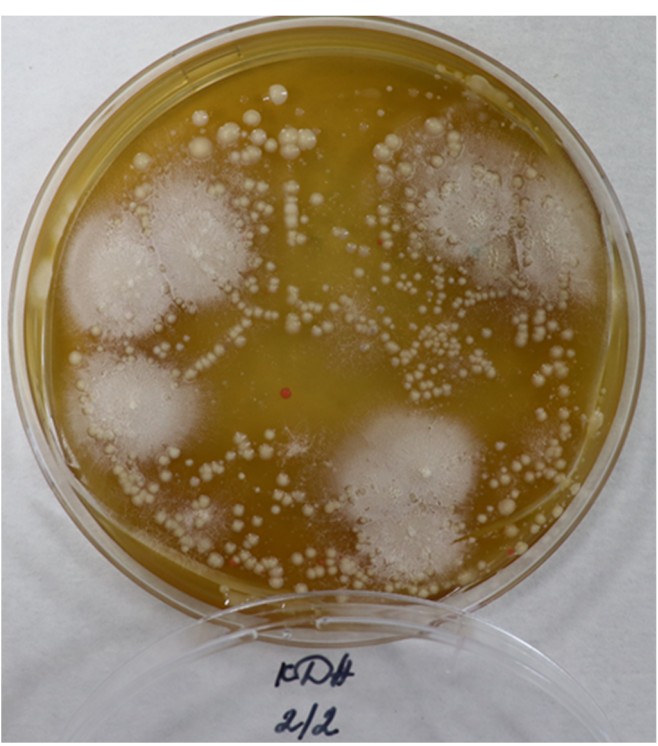

**Figure 4.** Colonies of filamentous fungi and yeast from a sample of water extract a (dilution 1:10$^2$).

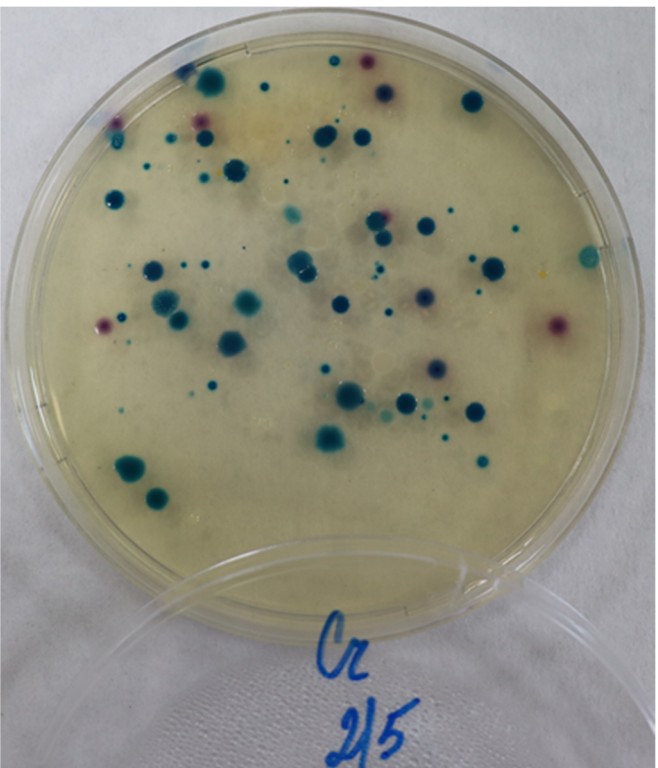

**Figure 5.** Bacteria of the *Escherichia coli* group on Chromagar Orientation growth medium, obtained from a sample of water extract 4 (dilution 1:10$^5$).

## 3. Discussion

Ten spot samples were studied, as well as a number of combined samples of municipal solid waste from the landfill of the city of Shymkent. The physical and chemical character-



istics were determined, and it was found that the average density of MSW is at a typical level for developed countries.

Food waste is represented mainly by the inedible remains of vegetables and fruits with signs of decay. Often, this type of waste is presented by tied plastic bags, which, combined with a high humidity and favorable temperatures in the warm season, contributes to the growth of microorganisms. It is worth noting that these wastes are the main source of unpleasant odors.

The low moisture content of the studied samples was caused by favorable climatic conditions, since the sampling was carried out during the warm season for this region, as well as by the storage of waste in the open air, which contributed to natural drying, since plastic waste is not a hygroscopic material. Due to hygroscopicity, paper waste is a source of moisture in the MSW samples.

An important parameter of any solid fuel is the ash content, the mass fraction of ash and the percentage of non-combustible (per anhydrous mass) residue, which is created from the mineral impurities of the fuel during its complete combustion. There is a difference between the external and internal ash content. The external ash content is the result of the fuel's foreign impurities; in the case of MSW samples, the main impurities are sand, dust and soil particles. Additionally, metal particles and (for example) aluminum foil from combined packages based on glued paper and cardboard will increase the ash content.

The average ash content is at a fairly high level, but most likely it is the surplus ash content, as evidenced by the composition of the ash, which has a high content of calcium, silicon, aluminum and magnesium, all of which are distinctive elements of the soil. By organizing separate collections in containers, as well as providing intermediate storage on the site, for example, a concrete site, the ash content can be significantly reduced when burning such fuel. At the same time, it is also necessary to minimize the release of ash particles into the atmosphere, which, according to the analysis, are very small in size and, accordingly, can be carried away with air currents.

Another important quality measure for fuel is the yield of volatile substances, which characterizes the quality of solid fuels and is taken into account when determining their rational industrial use. When heated without access to air, solid fuels decompose, releasing gas and vapor products, which are called volatile substances. Depending on the heating temperature, after the removal of volatile substances, a solid residue (kinglet), coke or semi-coke remains. Volatile substances are not contained in the free form in the fuel, but they are formed when heated, so we can only discuss their yield, not their content. The yield of volatile substances depends not only on the type of fuel, but also on the conditions of its heating. The composition of volatile substances includes valuable substances that are widely used in industry. Therefore, for example, volatile substances of coal contain benzene, toluene, ammonia, hydrogen, methane, etc. The volatile substances formed during the dry distillation of wood contain methane, carbon monoxide, acetic acid, methyl alcohol, etc.

The net calorific value is the most important indicator of the quality and energy properties of fuel and characterizes its value. The net calorific value differs essentially from the gross value only in that the water formed during the combustion of fuel does not condense but remains in the form of steam and is removed with flue gases. The net calorific value is lower than the gross value due to the heat of the condensation of steam, which is formed from the moisture of the fuel and hydrogen of the organic mass; this turns into water during combustion.

The net calorific value of the combustible fraction of municipal solid waste containing various organic materials is comparable to such values for charcoal and anthracite (~27 MJ/kg). At the same time, taking into account these parameters for paper and wood waste, it becomes clear that the main contribution to the calorific value is made by plastic waste.

The accurate determination of the content of macro-elements in solid fuel from municipal solid waste is necessary to solve environmental and technological problems both at the production stage and during combustion. The determination of macro-elements is

useful for the estimation of the behavior of ash during the combustion of solid fuels and the slagging of heating surfaces. The direct analysis of solid fuel from municipal solid waste is not possible due to insufficient sample homogeneity, which requires a preliminary decomposition.

A microbiological study showed that bacterial micro-flora predominated in the samples of water extracts. Significant amounts of bacteria of the *Escherichia coli* group were detected. Fungal micro-flora, represented by filamentous fungi and yeasts, was also found in the studied water samples. Of the detected micro-flora, the greatest danger to personnel involved in the processing and incineration of MSW is caused by bacteria of the *Escherichia coli* group, since virulent strains, when infected by the alimentary route, can cause gastroenteritis, peritonitis, sepsis, etc. Additionally, a high value of the coli index of samples (more than 1100) indicates a strong fecal contamination of the samples. These circumstances emphasize the need for workers to use personal protective equipment and for mandatory hygiene measures.

Therefore, the parameters of humidity, the ash content of volatile substances and the net calorific value are at the global average level.

## 4. Materials and Methods

Sampling was carried out in accordance with the "Sanitary and epidemiological requirements for the collection, use, application, disposal, transportation, storage and disposal of production and consumption waste" by order of the Minister of Health of the Republic of Kazakhstan, dated 23 April 2018 No. 187 [23]. Samples were taken in polyethylene hermetically sealed containers with a volume of 60 L, using a bayonet and square-faced shovels.

The morphological composition of municipal waste was determined according to the "Methodology for determining the morphological composition of municipal solid waste" by order of the Agency of the Republic of Kazakhstan for construction and housing and communal services, dated 10 January 2012 No. 4 [24].

Samples for further testing were prepared according to ISO 21645:2021 [25]. Two main methods were used in the sample preparation: sample weight reduction by separation and sample grinding [26]. The Table 9 lists the used research methods with justification of their necessity.

**Table 9.** Research methods.

| № | Research Method | Justification |
|---|---|---|
| 1 | ISO 21660-3:2021. Solid recovered fuels—Determination of moisture content using the oven dry method | Moisture in the fuel is ballast, you need to know if pre-drying is necessary. |
| 2 | ISO 21656:2021. Solid recovered fuels—Determination of ash content | The amount of ash characterizes the volume of generated solid waste which have to be disposed after fuel combustion. |
| 3 | ISO 22167:2021. Solid recovered fuels—Determination of content of volatile matter | A parameter taken into account when determining the rational industrial use of solid fuels. It also shows the amount of combustible gases that can be obtained during pyrolysis. |
| 4 | ISO 1928:2009. Solid mineral fuels—Determination of gross calorific value by the bomb calorimetric method and calculation of net calorific value | The most important indicator of fuel quality, characterizes its calorific value. |

## 5. Conclusions

In sum, the energy potential of municipal solid waste from the landfill of the city of Shymkent is shown. It has been established that organizing the separate collection of plastic and paper waste allows us to increase the calorific value of municipal solid waste and to reduce its average ash content during combustion.

Considering the amount of paper and plastic in the MSW of this region and how they can be recycled from an economical point of view is critical. Unfortunately, most of the MSW in this region is buried without any recycling process; therefore, the Shymkent municipal organization should make an instant and effective decision to recycle the MSW that is buried daily.

**Author Contributions:** Conceptualization, A.Z.A.; formal analysis, O.S.K.; investigation, I.E.A.; methodology, N.S.I.; project administration, A.Z.A.; resources, I.E.A.; visualization, O.S.K. and I.E.A.; writing—original draft, N.S.I.; writing—review and editing, A.Z.A. and N.S.I. All authors have read and agreed to the published version of the manuscript.

**Funding:** This research was funded by the Science Committee of the Ministry of Education and Science of the Republic of Kazakhstan, grant number AP09058076 "Research of energy potential and environmental safety of the municipal solid waste from Kazakhstani landfills".

**Institutional Review Board Statement:** Not applicable.

**Informed Consent Statement:** Not applicable.

**Data Availability Statement:** Not applicable.

**Conflicts of Interest:** The authors declare no conflict of interest. The funders had no role in the design of the study; in the collection, analyses, or interpretation of data; in the writing of the manuscript; or in the decision to publish the results.

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
