# Peer review of "Assessment of the Energetical and Biological Characteristics of Municipal Solid Waste from One of the Largest Landfills in Kazakhstan"

_recycling, doi:10.3390/recycling7050080_

Round 1
Reviewer 1 Report
This study presented the composition of waste from a landfill in Shymkent city, Kazakhstan. The manuscript was well-written and explained in detail. It might be easy to understand even if the readers are not from the related fields. The information from this manuscript will be important data for not only the people in Kazakhstan but also in the global to understand the situation of waste management as a case study. This manuscript should be accepted to be published in this journal after the minor revision. Please check my comments and suggestion below.
- The abstract is not well written. Most results were presented with many numbers but lack of discussion and conclusions. Why Shymkent city? Please make it clear.
- In the introduction parts, more specific backgrounds and objectives need to be explained.
- Generally, the materials and methods section should be placed before the results section. Please try to simplify. Also, the methods used for sample collection should be explained in more detail. How did you collect the samples? Do you have a specific point of collection or they were collected randomly?
- Table 1: It was explained the density of waste in this study was different compared with other countries. Why? This result needs to be discussed.
- Table 4 – 6: the abbreviations (i.e., No, Wa, Ad, and Vd) need to be described.
- Table 8: the reason for using only samples no.2,4, and 6 in the microbiological analysis should be explained.
Author Response
Пожалуйста, посмотрите приложение

Reviewer 2 Report
The Paper is quite up to the mark but it lacks certain grammatical mistakes like missing some punctuation. The paper is quite structured and below said corrections can be implemented for betterment: Q1. The more latest literature review can be added in the introduction part including their proper references. Q2. The Paper can be accepted although some sentences can be shortened and grammatical mistakes can be removed. Q3. More references can be added to support the result and discussion part in every aspect of influencing factors. Detailed comments are attached in a word file.

Round 2
Reviewer 2 Report
No further comments